# Simultaneous Predictions of Chemical and Phase Equilibria in Systems with an Esterification Reaction Using PC-SAFT

**DOI:** 10.3390/molecules28041768

**Published:** 2023-02-13

**Authors:** Moreno Ascani, Gabriele Sadowski, Christoph Held

**Affiliations:** Laboratory of Thermodynamics, Department of Biochemical and Chemical Engineering, TU Dortmund University, Emil-Figge Str. 70, 44277 Dortmund, Germany

**Keywords:** liquid–liquid equilibrium, reaction equilibrium, thermodynamics, equation of state, reactive separation

## Abstract

The study of chemical reactions in multiple liquid phase systems is becoming more and more relevant in industry and academia. The ability to predict combined chemical and phase equilibria is interesting from a scientific point of view but is also crucial to design innovative separation processes. In this work, an algorithm to perform the combined chemical and liquid–liquid phase equilibrium calculation was implemented in the PC-SAFT framework in order to predict the thermodynamic equilibrium behavior of two multicomponent esterification systems. Esterification reactions involve hydrophobic reacting agents and water, which might cause liquid–liquid phase separation along the reaction coordinate, especially if long-chain alcoholic reactants are used. As test systems, the two quaternary esterification systems starting from the reactants acetic acid + 1-pentanol and from the reactants acetic acid + 1-hexanol were chosen. It is known that both quaternary systems exhibit composition regions of overlapped chemical and liquid–liquid equilibrium. To the best of our knowledge, this is the first time that PC-SAFT was used to calculate simultaneous chemical and liquid–liquid equilibria. All the binary subsystems were studied prior to evaluating the predictive capability of PC-SAFT toward the simultaneous chemical equilibria and phase equilibria. Overall, PC-SAFT proved its excellent capabilities toward predicting chemical equilibrium composition in the homogeneous composition range of the investigated systems as well as liquid–liquid phase behavior. This study highlights the potential of a physical sound model to perform thermodynamic-based modeling of chemical reacting systems undergoing liquid–liquid phase separation.

## 1. Introduction

The study of chemical reactions in multiphase systems is important from a purely scientific point of view and becomes crucial in technical applications involving chemical and phase equilibria (CPE) [1]. The presence of one or more chemical reactions adds internal degrees of freedom (i.e., the species cannot only move between phases but also be converted into different components) to the multicomponent system that must be considered for designing chemical processes, thus increasing the intrinsic complexity of the system. Even for kinetically controlled systems, CPE dictates the ultimate state of the system at the given conditions ($T,p,\bar{x}$) [2,3,4] since the molecules that take part in the reactive systems (i.e., the reacting agents) ultimately decide on the position of the reaction equilibrium and on the heterogeneity of the reaction system. Thermodynamic analysis, performed using accurate thermodynamic models, represents a powerful tool to enable the estimation of the reaction equilibrium (i.e., the maximum yield of a chemical reaction) and to assess the possible demixing into two or more phases over the reaction coordinate. The integration of chemical reaction and phase separation into one single unit, called reactive separation, has several applications in the industry and in academia, e.g., reactive distillation [5,6,7,8], reactive extraction [9,10,11], reactive crystallization [12,13] or reactive absorption [14,15,16]. Reactive processes may offer several advantages over their non-reactive counterparts, such as an increase in reaction yield and selectivity, the overcoming of thermodynamic restrictions (e.g., azeotropes), energy saving or capital cost reduction [17,18,19]. Further, reactions in multiple phases systems are encountered in living systems such as coacervates [20,21], in downstream processes for biomolecules’ purification [22,23] or in geological fluid systems [24]. In process engineering, several methods have been developed so far to generate feasible flowsheets for a given separation/reaction task or to dimension and optimize the required unit operations [17], with all the proposed methods relying on the availability of experimental data. While accurate thermodynamic and kinetic data are required for a rigorous apparatus design, more qualitative data reflecting the thermodynamic behavior of the system (such as the appearance of azeotropes or the presence of a heterogeneity at given conditions, see for instance [25]) must be known at the early stage of project development. In this sense, advanced thermodynamic models offer great potential for cost and time saving since they allow existing experimental data to be correlated and their values to be predicted at experimental conditions that were not investigated [26]. As well as this correlative/extrapolative purpose, other properties that are difficult to measure (such as the interface tension [27] or diffusion behavior [28]) can be estimated using a thermodynamic model.

Motivated by the aforementioned importance, reactive phase equilibria have been the subject of many scientific works, dealing with theory and/or with experimental studies. Several authors focused on a pure theoretical–mathematical description of the reactive phase equilibrium problem, ranging from exploring the conditions of uniqueness of the phase equilibrium solution and the chemical equilibrium (CE) solution [29,30] or on its properties [31,32,33] to the development of new mathematical formulations of the CPE problem [34,35,36,37,38,39,40]. Doherty and coworkers [32,33,41,42] proposed the use of transformed variables, a concept already developed in soviet literature [43,44], to represent reactive phase diagrams and formulate working equations for CPE calculations. Later, they adapted the transformed variable to concrete calculation problems, for instance, to phase stability analysis [45] and the prediction of reactive azeotropes in a reactive mixture [46,47]. Several algorithmic approaches to solve the combined CPE problem were developed [48,49,50,51,52,53,54,55,56,57]. One successful approach is, for instance, the RAND method by Gautam et al., based on the work of White et al. [58]. The RAND algorithm is nowadays employed in commercial software such as Aspen Plus [59]. Recently, Koulocheris et al. [60] published an algorithmic approach to perform CPE of reactive VLEs using traditional G^E^-models and tested their approach to several systems containing azeotropes. For an extensive discussion on the different available algorithms, we refer to the recent literature [53,61,62,63]. The group of Toikka has worked extensively in the field of CPE with theoretical considerations [64,65,66,67,68,69,70] and experimental/modeling contributions [71,72,73,74,75,76]. They have also published some extensive reviews on existing data for CPE systems [1,67].

The purpose of the present work is to test the thermodynamic model Perturbed Chain Statistical Associating Fluid Theory (PC-SAFT) for the modeling of CE in systems exhibiting liquid–liquid equilibria (LLE) and to develop an algorithmic approach to perform this task. PC-SAFT was chosen as it has already been used to model the CE in an esterification system of 1-butanol with acetic acid by Grob et al. [77], and by Riechert et al. [78] to model the reaction equilibrium in the esterification of ethanol and 1-propanol with acetic acid. The electrolyte version ePC-SAFT was also successfully used to model the CE of enzyme-catalyzed reactions [79,80], and the Michaelis constant [81,82,83], which can also be interpreted as a reaction equilibrium between free and bounded enzyme. However, none of these works considered the combined CPE problem. The proposed algorithm is a stoichiometric, equation-solving method based on a double-nested procedure with a successive update of the fugacity coefficients of all the components. The number of phases and initial composition estimates are provided by the tangent plane stability analysis [84,85]. Although the general idea of the double-nested procedure is widely known (see, for instance, [54,57,60,86,87,88]), in this work, we proposed some new ideas to improve the robustness of the CPE calculation. The algorithm was applied to predict the CPE of two quaternary systems with an esterification reaction and the modeling results were compared with experimental data from the literature. Section 2.1 provides a summary of the theory of chemical reactions in multiphase equilibria, the different approaches to treat the CPE and the consequences of the occurrence of chemical reactions on the topology of phase diagrams. Details about the derivation and the structure of the proposed algorithm can be found in Section 2.2. Section 3.1, Section 3.2, Section 3.3, Section 3.4 and Section 3.5 provide a description of the investigated systems and the modeling strategy, as well as the calculated phase diagrams including the reaction equilibria. Details about PC-SAFT are summarized in Appendix A.

## 2. Algorithmic Approach

### 2.1. Thermodynamics of Chemical Reactions and Multiple Liquid Phase Equilibria

For a system at given temperature *T*, pressure *p* and respective total moles ${\bar{n}}^{F}=\left(n_{1}^{F},n_{2}^{F},\ldots ,n_{N}^{F}\right)$ of each components i=1,…,N at an arbitrarily chosen feed composition *F*, the mathematical solution of the CPE problem is defined by the number of phases π and number of moles $n_{i}^{\left(j\right)}$ of each component i in each phase j that minimizes the total Gibbs energy of the system (Equation (1)) [4,89]:(1)$$\underset{\overset{═}{n}}{\min}G=\overset{\pi }{\underset{j=1}{\sum }}\overset{N}{\underset{i=1}{\sum }}n_{i}^{\left(j\right)}\mu_{i}^{\left(j\right)}=\overset{\pi }{\underset{j=1}{\sum }}{\bar{n}}^{\left(j\right)}{}^{T}\cdot {\bar{\mu }}^{\left(j\right)}$$

This is subject to the element conservation (Equation (2)), mass balance (Equation (3)) of each component, and non-negativity of the number of moles of each component (Equation (4)) [90].
(2)$$\overset{═}{A}\cdot {\bar{n}}^{F}-\bar{b}=\bar{0}\;$$
(3)$${\bar{n}}^{F}-\overset{\pi }{\underset{j=1}{\sum }}{\bar{n}}^{\left(j\right)}=\bar{0}$$
(4)$$n_{i}^{\left(j\right)}\geq 0\;i=1,\ldots ,N\;j=1,\ldots ,\pi$$

In Equation (2), $\overset{═}{A}$ is the component–element matrix, which has dimension $N^{E}\times N$ ($N^{E}$ being the number of elements that build up the N components) and whose *i*-column ${\left(a_{1i},a_{2i},\ldots ,a_{N^{E}i}\right)}^{T}$ contains the number of each element present in component *i*. The $N^{E}$-dimensional array $\bar{b}$ contains the number of moles of each element present in the system (which remains unchanged). However, among the $N^{E}$ linear equations defined by Equation (2), only $N^{C}\leq N^{E}$ equations, which are given by the rank of the matrix $\overset{═}{A}$, $N^{C}=rank(\overset{═}{A})$, are sufficient to uniquely represent the condition of element conservation. The difference $N^{R}=N^{E}-N^{C}$ is denoted number of key reactions in the literature.

Based on the employed strategy to solve the CPE problem, two classes of methods can be distinguished, called non-stoichiometric and stoichiometric [38,90]. Non-stoichiometric methods directly attempt at solving the optimization problem given by Equations (1)–(4), using for instance the Kuhn–Tucker necessary conditions for minimization. Stoichiometric methods avoid the element balance constraints (Equation (2)) by formulating the component balance as a linear combination of $N^{R}$ reaction coordinates $\lambda_{j}$, $j=1,\ldots ,N^{E}$ according to Equation (5) [60].
(5)$$\;n_{i}^{F}=n_{i,0}^{F}+\overset{N^{R}}{\underset{k=1}{\sum }}\nu_{ik}\cdot \lambda_{k}\;\;i=1,\ldots ,N\;{\bar{n}}^{F}={\bar{n}}_{0}^{F}+\overset{═}{\nu }\cdot \bar{\lambda }\;$$

Here, ${\bar{n}}_{0}^{F}$ is a set of molar fractions satisfying the elemental abundance condition (Equation (2)) and $\overset{═}{\nu }$ being the stoichiometric matrix, which is a matrix of real numbers of dimension $N\times N^{R}$ satisfying the condition given by Equation (6) [90].
(6)$$\overset{═}{A}\cdot \overset{═}{\nu }=\overset{═}{0}\;$$

$\overset{═}{0}$ is a zero matrix of dimension $N^{E}\times N^{R}$. Rules for the correct choice of the number of key reactions $N^{R}$ and of the stoichiometric matrix $\overset{═}{\nu }$ is extensively discussed in the literature [3,91] and will not be repeated here. From Equations (1), (3) and (5), the necessary condition for the CPE can be derived, representing the condition that holds at equilibrium and is expressed by Equations (7) and (8) [34].
(7)$$\;\mu_{i}^{\left(1\right)}=\mu_{i}^{\left(2\right)}=\ldots .=\mu_{i}^{\left(\pi \right)}\;\;i=1,\ldots ,N$$
(8)$$\;\overset{═}{\nu }\cdot {\bar{\mu }}^{\left(R\right)}=\bar{0}\;$$
where ${\bar{\mu }}^{\left(R\right)}$ represents the array of the chemical potentials of all the components in one chosen reference phase. Equation (8) is equivalent to the analogue reformulation based on the activity-based equilibrium constant, which is written for a reaction *k* in Equation (9) [2,90].
(9)$$\overset{N}{\underset{i=1}{\sum }}\nu_{ik}\cdot \mu_{i}^{\left(R\right)}=\Delta^{R}g_{k}^{0}+RT\overset{N}{\underset{i=1}{\sum }}\ln{\left(x_{i}^{\left(R\right)}\gamma_{i}^{\left(R\right)}\right)}^{\nu_{ik}}=0$$

$\Delta^{R}g_{k}^{0}$ is the standard Gibbs energy of reaction. Equation (9) summarizes the well-known expression of the activity-based equilibrium constant $K_{a,k}$ given by Equation (10) [2,90].
(10)$$K_{a,k}=exp\left(-\frac{\Delta^{R}g_{k}^{0}}{RT}\right)=\overset{N}{\underset{i=1}{\prod }}{\left(x_{i}^{\left(R\right)}\gamma_{i}^{\left(R\right)}\right)}^{\nu_{ik}}\;K_{a,k}=K_{x,k}^{\left(R\right)}K_{\gamma ,k}^{\left(R\right)}$$

The previous conditions (Equations (7) and (8)), together with the mass balance (Equations (4) and (5)) build up a system of $\pi \cdot N+N^{R}$ equations that must be solved on $\pi \cdot N+N^{R}$ variables (which are the π·N
*non-negative* number of moles if each component in each phase and $N^{R}$ reaction coordinates). Equations (7) and (8) build the necessary equilibrium condition of an N-component multiphase system with $N_{R}$ key reactions. It must be pointed out that there could be more solutions of Equations (7) and (8) for a different number or even for the correct number of phases π (for instance, at the trivial solution or when an LL solution is present above the boiling point of the mixture) since the described condition is necessary but not sufficient.

For the determination of the equilibrium constant $K_{a}$ of a chemical reaction, two approaches can be employed: the first is to estimate the standard Gibbs energy of reaction $\Delta^{R}g_{k}^{0}$ from the standard energy of formation $\Delta^{F}g_{i}^{0}$ of each component *i* in a chosen reference state. The second approach relies on the availability of at least one experimental equilibrium composition ${\bar{x}}^{*,exp}$ and of a thermodynamic model to predict the set of activity coefficients $\bar{\gamma }\left(T,p,{\bar{x}}^{*,exp}\;\right)$, which can then be used to determine the $K_{a,k}$ based on Equation (10).

In a system with chemical reactions, the number of degrees of freedom *F* according to the Gibbs phase rule involving π phases and *N* components is reduced by the number of key reactions $N_{R}$, according to Equation (11) [64,92].
(11)$$F=2-\pi +N-N_{R}$$

The consequences of chemical reactions in a multiphase system for the resulting phase diagrams have been discussed extensively, for instance, in previous publications of the Toikka group [64,65,66]. Each key reaction decreases by one the dimension of the allowable composition space that can be reached by the system at equilibrium. For example, in a ternary homogeneous system with a key reaction of the form A+B⇌C at constant *T* and *p*, the number of degrees of freedom is F=2−1+3−1−2=1. That is, the dimension of the composition space is reduced from two (non-reactive) to one (reactive), and thus, all the equilibrium compositions will belong to a line, called chemical equilibrium curve (CE curve) [64]. The corresponding equilibrium composition of a quaternary system with one reaction A+B⇌C+D will span a chemical equilibrium surface (CE surface) [64]. A manifold of CE curves for different values of the equilibrium constant $K_{a}$ in a ternary ideal system are given in the ternary diagram of Figure 1.

If the system under consideration shows high non-ideality up to miscibility gap, some of the curves of the manifold can show a strong deviation from the ideal hyperbolic form shown in Figure 1. Othmer et al. [30] showed that for strongly non-ideal systems, up to three solutions can be present for some curves, although only one or two (belonging to a tie-line) corresponds to the stable solution. Figure 2 shows some CE curves, calculated for the same reaction A+B⇌C as of Figure 1, crossing the miscibility gap of a strongly non-ideal system. Calculations are performed using the algorithm developed in Section 2.2. The tie-line that connects two points at CE is called a reactive tie-line [32], with a further distinction of a unique reactive tie-line if it appears in a ternary system [1,32,64].

Whether or not phase split occurs in the reaction system is dictated by the properties of the pure components and of the mixture. The kind of molecules involved in the reaction mixture determine the equilibrium constant *K_a_*, which in turn dictates if the reaction path undergoes phase split(s). The ternary diagram (Figure 2) shows one CE curve passing through the homogeneous region (*K_a_* = 1.8), three crossing the two-phase region one time (one, however, showing one tangent point to the binodal more than crossing it) and one crossing the two-phase region twice (*K_a_* = 2.25). The composition points inside the binodal will not exist in a system in equilibrium and will split into two phases (given by two points connected by a tie-line). However, the crossing points of CE curve and binodal represents points belongs to the reactive tie-lines. An exemplary diagram of a reactive system with two reactive tie-lines is shown in Figure 3, representing the real solution of the system in Figure 2 with *K_a_* = 2.25.

### 2.2. Algorithm Architecture

In this section, our implementation of the algorithm to perform numerical calculation of the CPE is presented. Although the implementation is general, i.e., not limited to a single key reaction and neutral components (see, for instance, our previous work [93,94]), its scope in this work is to calculate the coupled reaction equilibrium and LLE of the two investigated esterification systems. Thus, only systems containing four molecular (i.e., non-charged) components and described by one key reaction are considered.

The phase equilibrium calculation starts with a given number of phases and an initial guess composition, which in our approach is provided by the tangent plan stability analysis [84,85]. Initially, a unimolar feed amount is initialized ($n_{i,0}^{F}=x_{i,0}^{F}$). The number of moles of each component in each phase is thus given by Equation (12).
(12)$$n_{i,0}^{\left(j\right)}=x_{i,0}^{\left(j\right)}\alpha_{0}^{\left(j\right)}$$

Instead of searching for a direct solution of Equation (7), the objective function is reformulated as given by Equation (13) for each component.
(13)$$1-\frac{x_{i}^{\left(j\right)}}{x_{i}^{\left(R_{i}\right)}}exp\left\{\ln\varphi_{i}^{\left(j\right)}-\ln\varphi_{i}^{\left(R_{i}\right)}\right\}=0\;\;\begin{array}{c} i=1,\ldots ,N_{neut}\;\\ j=1,\ldots ,\pi \;;\;j\neq R_{i} \end{array}$$

In Equation (13), $\varphi_{i}^{\left(j\right)}$ is the fugacity coefficient of component *i* in phase *(j)* and *R_i_* represents a reference phase, which is chosen for component *i*. Thus, $exp\left\{\ln\varphi_{i}^{\left(j\right)}-\ln\varphi_{i}^{\left(R_{i}\right)}\right\}$ represents the partition coefficient of component *i* between phase *j* and the reference phase *R_i_*. For a reactive system with $N^{R}$ key reactions, $N^{R}$ further equations must be formulated, representing the CE condition given by Equation (8). In our work, the $N^{R}$ objective functions are reformulated by Equation (14).
(14)$$1-\frac{K_{\gamma ,m}^{\left(R\right)}K_{x,m}^{\left(R\right)}}{K_{a,m}}=0$$

The superscript *(R)* in Equation (14) means that the activity coefficients in $K_{\gamma ,m}^{\left(R\right)}$ and the concentrations in $K_{x,m}^{\left(R\right)}$ refer to the component in only one reference phase (which do not have to be necessarily the same for all the components). Equation (13) builds up a set of $\left(\pi -1\right)\cdot N$ equations, and if $N^{R}$ key reactions must be defined, the total number of equations becomes $\left(\pi -1\right)\cdot N+N^{R}$.

Thus, the roots of the system of equations given by Equations (13) and (14) can be found by changing the value of $\left(\pi -1\right)\cdot N+N^{R}$ variables. For each component i, a reference phase $R_{i}$ is defined, which at the same time is used to impose the isofugacity criterion by Equation (13). The reference phase of each component is chosen as the phase in which the highest initial number of moles of component i (according to Equation (11)) is present at the beginning of the CPE calculation. The number of moles of each component i in the respective reference phase $R_{i}$ is found by mass balance by imposing that the total number of moles must be equal to the number of moles of the feed (as given by Equation (3)). If $N^{R}$ key reaction must be defined, then the total feed composition is corrected using the stoichiometric coefficients and $N^{R}$ reaction coordinates, which are varied by the algorithm as well. The implemented equation for the number of moles of each component in its reference phase is given by Equation (15).
(15)$$n_{i}^{\left(R\right)}=\left(n_{i}^{F}-\sum_{m=1}^{N_{R}}\nu_{i}^{\left(m\right)}\lambda_{m}\right)-\sum_{\begin{array}{c} j=1\\ j\neq R \end{array}}^{\pi }n_{i}^{\left(j\right)}$$

Thus, $\left(\pi -1\right)\cdot N+N^{R}$ variables ($\left(\pi -1\right)\cdot N$ number of moles $n_{i}^{\left(j\right)}$ of each component and $N^{R}$ reaction coordinates $\lambda_{m}$, $m=1,\ldots ,N^{R}$) are varied to find the root of the system of $\left(\pi -1\right)\cdot N+N^{R}$ equations given by Equations (13) and (14).

As noted by other authors, the computationally most expensive step in phase equilibrium calculation is the evaluation of the fugacity coefficients in Equations (13) and (14) [54,87,88]. Boston et al. [87] suggested to solve the working equations in an inner loop using constant values of the fugacity coefficients and to update them, after convergence, in an outer loop. The iteration was performed until the relative difference between two successive solutions fell below a certain value. Upon using this method in our algorithm, we found convergence for the investigated esterification systems in this study. However, we observed oscillation and ultimately divergence when treating systems of concentrated electrolytes, high-pressure VLE with supercritical components and concentrated polymer solutions. In order to guarantee general robustness, we modified the double-nested approach. Instead of working with constant fugacity coefficients in the inner loop, we approximated them as linear functions of the compositions using partial derivatives, as given by Equation (16).
(16)$$\ln\varphi_{i}^{\left(j\right)}\left(T,p,{\bar{x}}^{\left(j\right)}\right)=\ln\varphi_{i}^{\left(j\right)}\left(T,p,{\bar{x}}_{0}^{\left(j\right)}\right)+\overset{N}{\underset{k=1}{\sum }}\frac{\partial \ln\varphi_{i}^{\left(j\right)}}{\partial x_{k}}\left(x_{k,0}^{\left(j\right)}-x_{k}^{\left(j\right)}\right)\;\;\begin{array}{c} i=1,\ldots ,N\\ j=1,\ldots ,\pi \end{array}$$

The partial derivatives of the fugacity coefficients in Equation (16) were evaluated numerically by finite difference at the beginning of the calculation, and then updated from the previous values using a Broyden estimation [95] after each calculation step. The resulting non-linear system of equations was solved using a Newton algorithm with variable step length α (Equation (17)).
(17)$${\bar{X}}_{k+1}={\bar{X}}_{k}-\alpha \cdot {\overset{═}{J}}^{-1}\left({\bar{X}}_{k}\right)\cdot \bar{F}\left({\bar{X}}_{k}\right)$$

In Equation (17), $\bar{X}$ represents the array of $\left(\pi -1\right)\cdot N+N^{R}$ variables, $\bar{F}\left({\bar{X}}_{k}\right)$ represents the array of objective functions (Equations (13) and (14)) calculated at the point ${\bar{X}}_{k}$ and $\overset{═}{J}\left({\bar{X}}_{k}\right)$ is the respective Jacobian matrix. The partial derivatives in the Jacobian matrix are calculated via automatic differentiation using dual numbers [96]. The step length α is reduced if the new estimate ${\bar{X}}_{k+1}$ leads to one or more negative concentrations, or if the norm of the new objective function array is greater than the previous (i.e., if $\bar{F}\left({\bar{X}}_{k+1}\right)>\bar{F}\left({\bar{X}}_{k}\right)$). After convergence of Equations (13) and (14), the value of the fugacity coefficients and their derivatives in Equation (16) is updated and the iteration of Equation (17) is started again. The double-nested procedure is repeated until the change in the calculated mole numbers between two updates falls below a tolerance $\delta_{tol}=10^{-8}$.

### Algorithmic Structure

In following, the calculation procedure is shown using the hypothetical mixture of Figure 2 and Figure 3 as test systems. These systems serve also as a first validation of our approach, since they show characteristic topologies of reactive phase diagrams already reported in the literature [30,66]. In sum, the algorithmic procedure to perform a reactive flash calculation according to our strategy consists of the following steps:

1-First the feed composition ${\bar{x}}^{F}$, the temperature *p* and the pressure *T* is given. Initially, an homogeneous CE calculation is performed at these conditions, according to the stoichiometry of the defined key reactions $N^{R}$. This is equivalent to moving the composition point, along a trajectory imposed by the stoichiometry called stoichiometry line, to the (hyper-)surface (composition ${\bar{x}}^{F\prime }$) where the CE condition for each key reaction is fulfilled (Equation (8)). For a simple reaction A+B⇌C and the corresponding ternary phase diagram, this chemical equilibration step can be visualized in Figure 4.2-Secondly, phase stability analysis according to the (non-reactive) tangent plane distance function [84,85] is performed for the chemically equilibrated feed. If the equilibrated feed lies inside the miscibility gap, two estimates of both liquid phase concentrations are provided (Figure 5).3-Third, CE is performed for each of the single phases provided by step 2. This is equivalent to moving each single phase, according to the reaction stoichiometry, to the chemical equilibrium (hyper-)surface. The overall feed composition will move as well; however, it will in general not lie to the chemical equilibrium (hyper-)surface as with the single phases. This third step will finally provide good initial point for the final reactive flash calculation.4-Finally, rigorous reactive flash calculation according to the strategy proposed in the last section is applied. After final convergence, two equilibrium points that satisfy Equations (7) and (8) are returned (Figure 6).

Figure 7 summarizes, in a flowchart, the computational steps of the CPE procedure explained in this section and their calling order within the implemented algorithm.

## 3. Results

### 3.1. The Reaction Systems Considered in This Work

In this work, two esterification systems were considered to test our approach and the performance of PC-SAFT to predict, simultaneously, the occurrence of LLE along the CE. Those are the quaternary system acetic acid + 1-pentanol + pentyl acetate + water (system 1) and the system acetic acid + 1-hexanol + hexyl acetate + water (system 2) according to the chemical Equations (18) and (19).
(18)$${\mathrm{CH}}_{3}\mathrm{COOH}+C_{5}H_{11}\mathrm{OH}⇌{\mathrm{CH}}_{3}{\mathrm{COOC}}_{5}H_{11}+H_{2}O$$
(19)$${\mathrm{CH}}_{3}\mathrm{COOH}+C_{6}H_{13}\mathrm{OH}⇌{\mathrm{CH}}_{3}{\mathrm{COOC}}_{6}H_{13}+H_{2}O$$

The first system was characterized by Senina et al. [72] at *T* = 318.15 K and *p* = 1.013 bar, the second system was extensively studied by Schmitt et al. [97,98,99] in a larger temperature range (293.15–403.15K with special focus on the range 353.15–393.15K) at *p* = 1.013 bar. Within the investigated temperature and pressure range, all the binary subsystems given by the alcohols and respective acetate esters with water show partial miscibility [100,101]. Thus, both the quaternary and all the ternary subsystems show a miscibility gap [98,102,103]; the only exception is the homogenous system acetic acid + alcohol (1-pentanol or 1-hexanol) + ester (pentyl acetate or hexyl acetate). Esterification is a catalytic process and is practically frozen without a catalyst [72]: Schmitt [97] investigated the autocatalytic esterification of 1-hexanol with acetic acid at 298.15 K, showing that the reaction did not approach the equilibrium even after weeks. The reaction can be carried out in the presence of an inorganic acid (homogeneous catalysis) or using a solid catalyst (heterogeneous catalysis). Senina et al. [72] used HCl_aq_ in concentrations less than 2 wt%, whereas Schmitt [97] employed an ion-exchange resin (Amberlyst CSP2). Due to the relatively low catalyst concentration, the catalyst was not considered in our calculations since it only marginally affects the phase equilibrium. In both works [72,97], the measurement of the final equilibrium composition (homogeneous CE, LLE or simultaneous CE and LLE) was carried out via gas chromatography.

### 3.2. PC-SAFT Parameters for the Considered Reaction Systems

In order to model the subsystems of the esterification systems (18) and (19), the parameters of the applied model must be determined. All the pure-component PC-SAFT parameters used in this work were retrieved from the literature and are listed in Table 1.

The binary interaction parameters used in this work were in part retrieved from the literature and, if not available, were regressed from mixture properties (LLE data in binary or ternary systems and VLE data in binary systems, see Table 2).

For the calculation of the CE curves in Figure 2 and Figure 3, pure-component parameters listed in Table 3 were used. The hypothetical components are called A, B, C, as used in the calculated ternary diagrams, and the used hypothetical binary interaction parameters were chosen to: $k_{AB}$ = −0.045, $k_{BC}$ = −0.025, $k_{AC}$ = 0.045.

### 3.3. The Reaction Equilibrium Constants K_a_ of the Considered Chemical Reactions

For the determination of the equilibrium constant $K_{a}$ of both chemical reactions (Equations (18) and (19)), we used one experimental equilibrium composition ${\bar{x}}^{*,exp}$ and predicted a set of activity coefficients $\bar{\gamma }\left(T,p,{\bar{x}}^{*,exp}\;\right)$ using PC-SAFT and the parameters applied in Table 1 and Table 2. This was then used to determine the $K_{a,k}$ based on Equation (10). This method circumvents the approximations made in the estimation of the standard energy of formations. Senina et al. [72] measured the CE composition in the homogeneous region of system 1 as well as nine quaternary tie-lines at the CE at the given *T* and *p* conditions. Other experimental data [110,111] were determined in the homogeneous liquid phase but at saturation condition, i.e., along the condition of liquid–vapor coexistence. Since *T* and *p* at saturation vary continuously with composition, only the data of Senina et al. (determined at fixed *T* and *p*) were considered in this work. For the same reasons, only CE compositions at fixed *T* and *p*, determined by Schmitt et al. [97], were considered in this work for the determination of $K_{a}$ in system 2. The resulting *K_a_* values determined based on the experimental data from the literature are listed in Table 4.

### 3.4. Prediction Results of the CPE Problem for Both Reactions under Study

Figure 8 and Figure 9 show the CE surface in the composition tetrahedron of both systems, including the heterogeneous region of CE (“unique chemical reactive surface”, according to [72]). These results were obtained using our developed algorithm (Section 2.2) fed by PC-SAFT (see the Appendix A) and the used parameters (Table 1 and Table 2) for the activity coefficients as well as the equilibrium constants (Table 4). Both, the reaction surface (CE surface) and the liquid–liquid miscibility gap (binodal curve) were predicted in good agreement with the experimental data.

### 3.5. Discussion

Figure 8 and Figure 9 show that the CE composition is predicted quantitatively correct by PC-SAFT, for both systems at the investigated *T* and *p* conditions, and in the whole composition range. System 2 shows a much broader miscibility gap in the CE surface than system 1, even at a higher temperature (353.6 K, compared to 318.15 K of system 1). This is in accordance with the subsystems, i.e., the much greater miscibility gap of 1-hexanol and hexyl acetate with water compared to their homologues 1-pentanol and pentyl acetate. The absence of experimental data of the CE tie-lines for system 2 did not allow for a direct comparison with the two-phase CPE prediction results in this system. Experimental CE tie-lines are available for system 1, and thus they were compared with the PC-SAFT predictions. The predicted CE tie-lines show qualitative agreement with the experimental data. It can be observed from Figure 9 that deviations between PC-SAFT and the experiments occur when acetic acid is present in the system. This inaccurate behavior of PC-SAFT at high acetic acid concentrations is already knows from previous work [77] and is probably due to the lack of representation of the dimerization behavior of acetic acid and the cross-association with the other components present in the mixture. A more detailed investigation of phase equilibria with acetic acid should be carried out in the future, trying to better capture the real association behavior of acetic acid in complex mixtures. This may require the investigation of the more refined (and likely more phenomenological) parametrization strategies of acetic acid and binary mixtures containing acetic acid.

Nevertheless, in sum, it can be concluded that the mathematical algorithm that has been developed in this work allows a satisfying estimation of chemical equilibria as well as liquid–liquid phase separation in the chemical reaction space by using PC-SAFT as the input tool for the activity coefficients. The results shown for the CE and CPE of the quaternary esterification systems are pure predictions since the model was parametrized using only pure-component vapor pressure and density (to determine the pure-component parameters, see Table 1) as well as the VLE and LLE of the binary and ternary subsystems (to determine the binary interaction parameters in Equation (A5), see Table 2). This is an important contribution to the design of reactive systems that may undergo phase separation.

## 4. Conclusions

In this work, an algorithm was successfully designed and implemented to predict CPE in multiphase multicomponent systems. New ideas were proposed to improve the robustness of the calculation procedure when calculating the CPE of strongly non-ideal systems. The algorithm uses PC-SAFT to describe the thermodynamic behavior of the system, i.e., the fugacity coefficients of the reacting agents. Prior to modeling, the related literature on the thermodynamics of multiphase reactive systems was reviewed, and the proposed algorithm was tested against a hypothetical ternary mixture with a chemical reaction, showing that the topologies of reactive phase diagrams that are reported in the literature are also predicted well by our approach. Using the implemented algorithm, the predictive capability of PC-SAFT on the CPE could be tested successfully for the first time, against the simultaneous CE and LLE in two quaternary esterification systems, formed respectively by esterification of acetic acid and 1-pentanol and of acetic acid and 1-hexanol. The CE composition in the homogeneous phase were predicted quantitatively correct by PC-SAFT in both systems and over the whole composition range and the investigated *T* and *p* condition. The prediction of simultaneous CE and LLE was qualitatively correct in the whole composition range, showing higher deviations from experimental data in the presence of acetic acid. This study suggests potential improvements, possibly in a new parametrization strategy for pure and binary mixtures of acetic acid, but more importantly suggests the use of PC-SAFT to design reactive systems that may undergo phase separation.

## Figures and Tables

**Figure 1 molecules-28-01768-f001:**
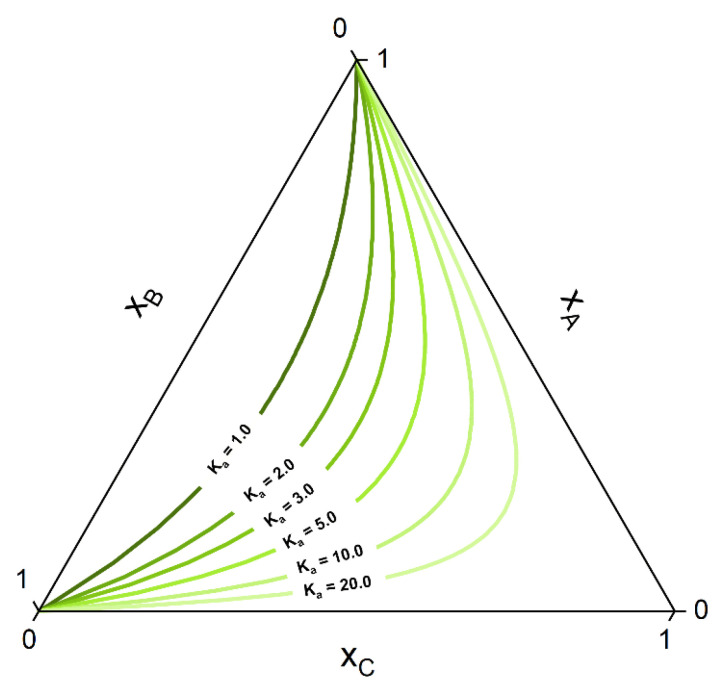
CE curves for different values of the CE constant K_a_ in a ternary system A+B⇌C that assumes ideal mixing behavior (K_γ_ = 1 in Equation (10)).

**Figure 2 molecules-28-01768-f002:**
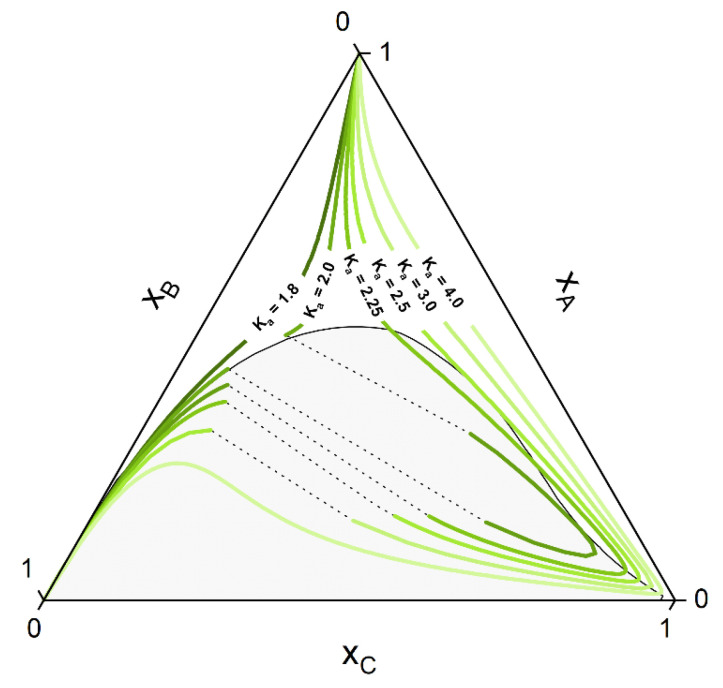
Hypothetical CE curves for different values of the CE constant K_a_ in a strongly non-ideal ternary system A+B⇌C with a miscibility gap. Calculations were performed using PC-SAFT with the algorithm developed in Section 2.2 and parameters listed in Section 3.2.

**Figure 3 molecules-28-01768-f003:**
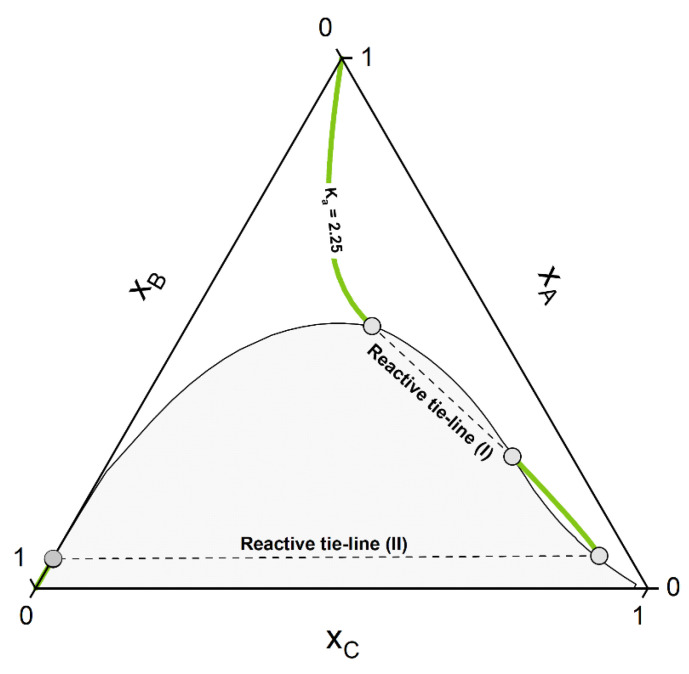
Resulting phase diagram of the hypothetical non-ideal ternary system A+B⇌C from Figure 2 for K_a_ = 2.25. Visible are the two disconnected CE curves passing through the homogeneous phase and the two unique reactive tie-lines I and II.

**Figure 4 molecules-28-01768-f004:**
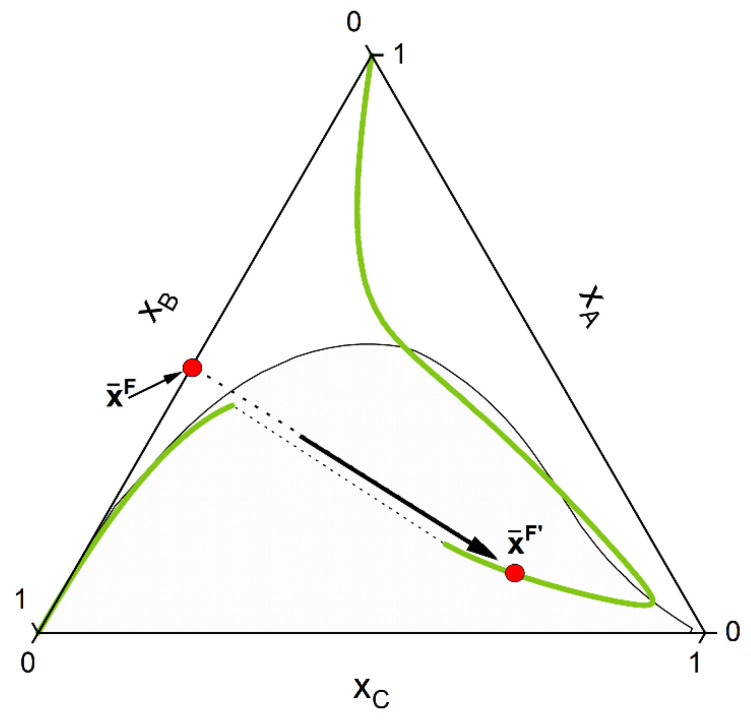
The initial feed point ${\bar{x}}^{F}$ is moved along a stoichiometry line to the CE curve (green line). In the depicted system the final composition ${\bar{x}}^{F\prime }$ lies inside the miscibility gap (grey area) for the chosen initial feed composition.

**Figure 5 molecules-28-01768-f005:**
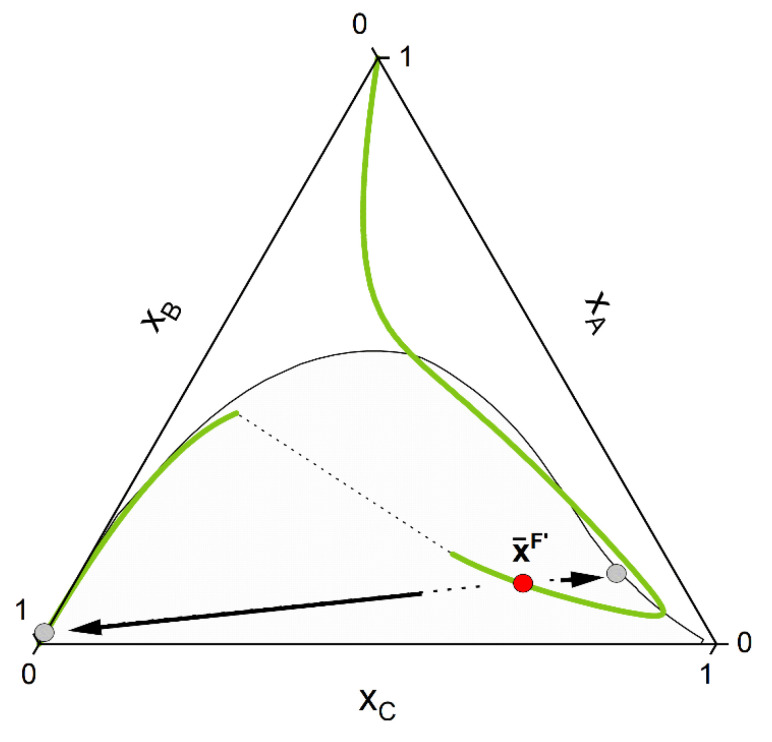
After reaching the CE curve, phase stability is performed for the chemical equilibrated feed ${\bar{x}}^{F\prime }$. The tangent plane criterion is implemented in this work, which provides estimate of the phase compositions (grey points) if instability of the liquid phase is detected.

**Figure 6 molecules-28-01768-f006:**
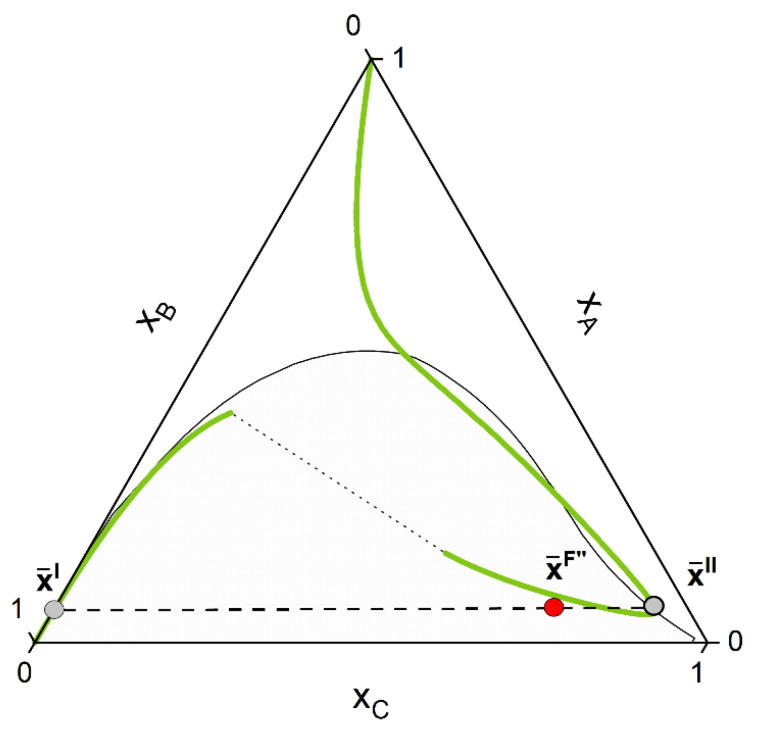
After convergence of the CPE calculation, the concentration of the two phases ${\bar{x}}^{\prime }$ and ${\bar{x}}^{\prime \prime }$ belonging to the same tie-line and the chemical equilibrium curve (the so-called “unique reactive tie-line”) is found. The resulting feed composition still remains on the stoichiometric line (NOT on the chemical equilibrium line).

**Figure 7 molecules-28-01768-f007:**
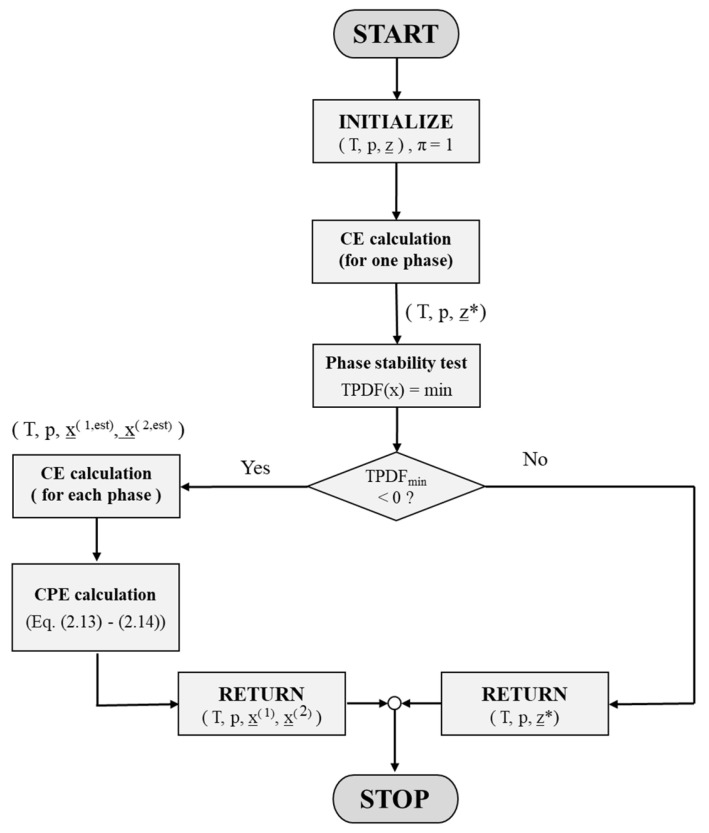
Flowchart that illustrates the computational steps of the CPE procedure implemented in the proposed algorithm. z* denotes a chemical equilibrated, homogeneous feed (relevant for a feed lying outside the miscibility gap), and x^(1,est)^ and x^(2,est)^ denote composition estimates of the heterogeneous feed after the phase stability test.

**Figure 8 molecules-28-01768-f008:**
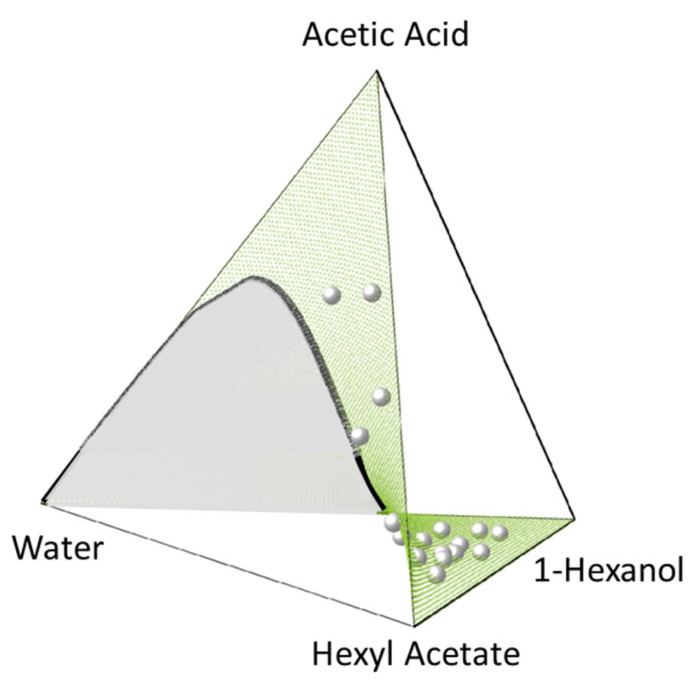
Quaternary phase diagram of system 2 (Equation (19)) at 353.6 K and 1 bar showing the PC-SAFT-predicted CE surface (green surface) and the PC-SAFT-predicted binodal (black curve encompassing the grey area). Experimental CE compositions of Schmitt et al. [97] are represented as grey spheres. All PC-SAFT predictions using parameters in Table 1 and Table 2.

**Figure 9 molecules-28-01768-f009:**
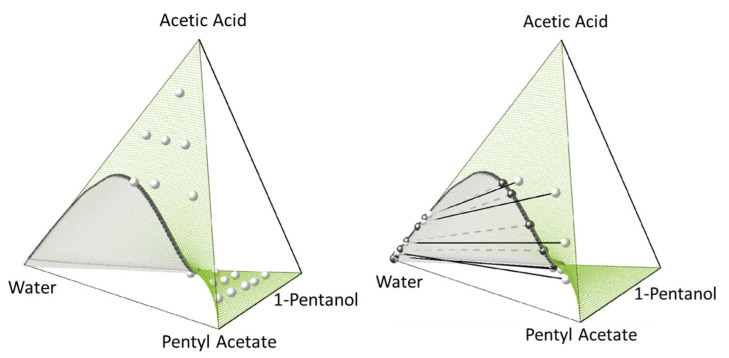
Quaternary phase diagrams of system 1 (Equation (18)) at 318.15 K and 1 bar showing the PC-SAFT-predicted CE surface (green surface) and the calculated binodal (black curve encompassing the grey area). **Left**: Experimental CE compositions of Senina et al. [72] (grey spheres). **Right**: Experimental tie-lines [72] (black spheres connected by a dashed line) and PC-SAFT-predicted tie-lines (grey spheres connected by a solid line). All PC-SAFT predictions using parameters in Table 1 and Table 2.

**Table 1 molecules-28-01768-t001:** PC-SAFT pure-component parameters used in this work to model the CE and LLE in the investigated systems.

Component	$m_{i}^{seg}/-$	$\sigma_{i}/Å$	$u_{i}k_{B}^{-1}/K$	$N_{i}$	$\varepsilon^{A_{i}B_{i}}k_{B}^{-1}/K$	$\kappa^{A_{i}B_{i}}/\text{-}$	Ref.
Water	1.2047	*	353.95	2	2425.7	0.04509	[104]
Acetic acid	1.3402	3.8582	311.59	2	3044.4	0.07555	[105]
1-Pentanol	3.6260	3.4508	247.28	2	2252.1	0.01033	[105]
1-Hexanol	3.5146	3.6735	262.32	2	2538.9	0.00575	[105]
Pentyl Acetate	4.7077	3.4729	234.57	2	0.0	0.04509	[106]
Hexyl Acetate	4.8847	3.5834	241.42	2	0.0	0.04509	[107]
$*\sigma =2.7927+\left(10.11e^{-0.01775\;T}-1.417e^{-0.01146\;T}\right)$							

**Table 2 molecules-28-01768-t002:** Binary interaction parameters used in this work to model multicomponent mixtures using PC-SAFT. Definition of the k_ij_ values according to the Appendix A.

Component 1	Component 2	*k_ij,_* _298.15_ */-*	*k_ij,T_*/*K*	Property Used for Estimation	Ref.
Water	Acetic acid	−0.1247	-	VLE-binary	[107]
Water	1-Pentanol	0.001604	0.00016	LLE-binary	[108]
Water	Pentyl Acetate	−0.0228	-	LLE-binary	This work (using data from [100])
Water	1-Hexanol	0.010105	0.000404	LLE-binary	[108]
Water	Hexyl Acetate	−0.01	0.0015	LLE-binary	This work (using data from [100])
Acetic acid	1-Pentanol	−0.1	-	LLE-ternary	This work (using data from [103])
Acetic acid	1-Hexanol	−0.033	-	LLE-ternary	This work (using data from [103])
Acetic acid	Pentyl Acetate	−0.1	-	LLE-ternary	This work (using data from [102])
Acetic acid	Hexyl Acetate	−0.08	−0.0004	LLE-ternary	This work (using data from [98])
1-Pentanol	Pentyl Acetate	−0.0095	-	VLE-binary	This work (using data from [109])
1-Hexanol	Hexyl Acetate	−0.0042	-	VLE-binary	This work (using data from [98])

**Table 3 molecules-28-01768-t003:** PC-SAFT pure-component parameters of hypothetical mixture A+B+C used to calculate the exemplary CE curves in Figure 2 and Figure 3.

Component	$m_{i}^{seg}/-$	$\sigma_{i}/Å$	$u_{i}k_{B}^{-1}/K$	$N_{i}$	$\varepsilon^{A_{i}B_{i}}k_{B}^{-1}/K$	$\kappa^{A_{i}B_{i}}/\text{-}$
A	2.4000	3.2000	200.00	2	2500.0	0.05
B	1.0800	3.0000	400.00	2	2500.0	0.05
C	2.8000	3.8000	280.00	0	-	-

**Table 4 molecules-28-01768-t004:** Obtained *K_a_* values for both systems 1 and 2, as well as the respective conditions (*T* and *p*) and the according references for the experimental equilibrium compositions.

System	T/K	p/bar	$K_{a}/-$	Ref. (for the Data)
1	318.15	1	43.99	[72]
2	353.6	1	22.92	[97]

## Data Availability

The data supporting the reported results are all given in this manuscript and in Appendix A.

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
