# Peer review of "Simultaneous Predictions of Chemical and Phase Equilibria in Systems with an Esterification Reaction Using PC-SAFT"

_molecules, 2023, doi:10.3390/molecules28041768_

Round 1
Reviewer 1 Report
The subject of manuscript is of some importance from theoretical point of view. It corresponds to Molecules. Mathematical algorithm developed by authors allows a satisfying estimation of chemical equilibria and liquid-liquid phase equilibria in chemically equilibrium states using PC-SAFT. The study suggests the use of PC-SAFT to design reactive systems with phase splitting. The article contains excellent illustrations that help to understand the features of the reaction in a heterogeneous region.
I would recommend that the authors check the punctuation (row 33, for example) and possible typos in the text of the article, after which the article can be published in the form in which it is.
Author Response
Answer Thanks to the reviewer for the suggestion. We substantially rechecked the manuscript, paying attention to error and better formulations, in agreement also to the comment of the other reviewers.Answer Thanks to the reviewer for the suggestion. We substantially rechecked the manuscript, paying attention to error and better formulations, in agreement also to the comment of the other reviewers.
Reviewer 2 Report
The authors of the manuscript “Simultaneous predictions of chemical and phase equilibria in systems with an esterification reaction using PC-SAFT” studied an algorithm which is designed and implemented in order to predict CPE in multiphase multicomponent systems. The algorithm use PC-SAFT to describe the thermodynamic behavior of the system. Using the implemented algorithm, the predictive capability of PC-SAFT on CPE could be tested successfully for the first time, against the simultaneous CE and LLE in two quaternary esterification systems. It can be published in “Molecules” after major revision. The concerns which should be considered are as follows:
1. The introduction part is very confusing and it is not clear what the authors trying to express, so we suggest that the authors should strengthen the logic of the introduction. And it lacks the development and background of the algorithm. It is recommended that the authors could succinct and supplement the relevant content respectively.
2. The model of the binary interaction parameter based on should be point out. Please verify the model.
3. It is mentioned in the manuscript that “No adjustable parameters were fit to the reactive system, all parameters were fit to the properties of the molecules and the phase behavior of some non-reactive mixtures.” Please explain the meaning of this sentence.
4. It is mentioned in the manuscript that “A more detailed investigation of phase equilibria with acetic acid should be carried out in the future, and possibly further parametrization strategies of acetic acid should be investigated”. I do not understand what the meaning of this sentence is to this study. Perhaps, you should add future outlook to this manuscript.
5. The abstract and conclusion in the manuscript do not reflect the innovation of the manuscript and the content is messy. Please refine the languages and rewrite these contents. Though the performance of the modified algorithmic is improved, the novelty of the manuscript should be emphasized.
6. The structure of the manuscript “1. Introduction 2. Results 3. Discussion 4. Materials and methods 5. Conclusion” seems unreasonable. Please modify the structure of the manuscript according to the content to make the manuscript more logical.
7. The format of references in the manuscript is inconsistent, such as: 11. Schulz R, Waluga T. 7 Reactive extraction. Process Intensification: by Reactive and Membrane-assisted Separations 2022:363. 14. Kenig EY, Schneider R, Górak A. Reactive absorption: Optimal process design via optimal modelling. Chemical Engineering Science 2001; 56(2):343–50. Please follow the instructions of the “Author's Guide” about the journal to make a careful review and revision of the references
8. The developed algorithm within this work was applied to predict CPE of two quaternary systems with an esterification reaction. This prediction method is very good and the experimental method has been improved. You can refer to the following literatures, which are about the determination of liquid-liquid equilibrium and can provide experimental and theoretical support for the manuscript. (Journal of Solution Chemistry, 2019, 48: 1547–1563; Fluid Phase Equilibria, 2016, 412: 205-210; Separation and Purification Technology, 2020, 247: 116937) You can refer to the following literatures for the study of PC-SAFT in liquid-liquid equilibrium. (Industrial & Engineering Chemistry Research, 2020, 59(18): 8836–8847; Fluid Phase Equilibria, 2011, 302(1-2): 169-178; Polymer, 2016, 104: 149-155; Fluid Phase Equilibria, 2020, 523, 112689)
Author Response
- The introduction part is very confusing and it is not clear what the authors trying to express, so we suggest that the authors should strengthen the logic of the introduction. And it lacks the development and background of the algorithm. It is recommended that the authors could succinct and supplement the relevant content respectively.
Answer Thanks to the reviewer for the comment. We extended some arguments in the main text and given a more concise logic. We also thank the reviewer for the literature suggestion in point 8, which we used to strength our arguments in the introduction. Also we extended and moved the background of the algorithm from the Appendix into the main text and we framed it within the scope of the paper.
- The model of the binary interaction parameter based on should be point out. Please verify the model.
Answer Thanks for the comment. There is no specific model of the binary interaction parameters. The only model used in this work is PC-SAFT, which requires pure-component parameters and one binary interaction parameter (called kij). The corresponding Equation is shown in the Appendix (Eq. A.5).
- It is mentioned in the manuscript that “No adjustable parameters were fit to the reactive system, all parameters were fit to the properties of the molecules and the phase behavior of some non-reactive mixtures.” Please explain the meaning of this sentence.
Answer The showed results (quaternary CE and CPE) are pure prediction since, for the parametrization of the model, only density and vapor pressure of the pure components as well as phase equilibria (VLE and LLE) of binary and ternary subsystems were used. In other words, the model never saw the phase equilibria (CE or CPE) of the 4-component system against which the calculations were tested. We modified the corresponding sentences in the discussion accordingly. Thanks to the reviewer for the comment.
- It is mentioned in the manuscript that “A more detailed investigation of phase equilibria with acetic acid should be carried out in the future, and possibly further parametrization strategies of acetic acid should be investigated”. I do not understand what the meaning of this sentence is to this study. Perhaps, you should add future outlook to this manuscript.
Answer The sentence was actually confusing, we basically propose to address the specific thermodynamic behavior of acetic acid and mixtures of it, trying to capture the phenomenology of the system (i.e. the complex association behavior of carboxylic acids). Possibly a new parametrization strategy may be required for this. In this version of the paper, we refined the outlook and the discussion about the difference between modeled and experimental data and given a possible explanation of the observed deviations. We think the discussion section will be much clearer now. Thank you for the suggestion
- The abstract and conclusion in the manuscript do not reflect the innovation of the manuscript and the content is messy. Please refine the languages and rewrite these contents. Though the performance of the modified algorithmic is improved, the novelty of the manuscript should be emphasized.
Answer Thanks for the comment. We extended and refined the abstract and conclusion, and we also pointed out the novelty of this work, i.e. the algorithm approach and the comparison of PC-SAFT toward CPE for the first time.
- The structure of the manuscript “1. Introduction 2. Results 3. Discussion 4. Materials and methods 5. Conclusion” seems unreasonable. Please modify the structure of the manuscript according to the content to make the manuscript more logical.
Answer We agree to the reviewer about the need of a more appropriate structure. We re-ordered the sections into: 1. Introduction 2. Algorithmic approach 3. Results 4. Conclusion, and the respective subsections.
- The format of references in the manuscript is inconsistent, such as: 11. Schulz R, Waluga T. 7 Reactive extraction. Process Intensification: by Reactive and Membrane-assisted Separations 2022:363. 14. Kenig EY, Schneider R, Górak A. Reactive absorption: Optimal process design via optimal modelling. Chemical Engineering Science 2001; 56(2):343–50. Please follow the instructions of the “Author's Guide” about the journal to make a careful review and revision of the references
Answer We checked our reference and made the format of papers / books / book chapters consistent. Thank you for the comment on that.
- The developed algorithm within this work was applied to predict CPE of two quaternary systems with an esterification reaction. This prediction method is very good and the experimental method has been improved. You can refer to the following literatures, which are about the determination of liquid-liquid equilibrium and can provide experimental and theoretical support for the manuscript. (Journal of Solution Chemistry, 2019, 48: 1547–1563; Fluid Phase Equilibria, 2016, 412: 205-210; Separation and Purification Technology, 2020, 247: 116937) You can refer to the following literatures for the study of PC-SAFT in liquid-liquid equilibrium. (Industrial & Engineering Chemistry Research, 2020, 59(18): 8836–8847; Fluid Phase Equilibria, 2011, 302(1-2): 169-178; Polymer, 2016, 104: 149-155; Fluid Phase Equilibria, 2020, 523, 112689)
Answer Thanks to the reviewer for suggesting further literature. We could not find a section in the paper where to refer to the first four literature, since our work is not about measuring the liquid-liquid equilibrium but on developing an algorithm to compute the CPE and testing the capability of PC-SAFT in doing that. But we found the last four literatures very useful for our discussion and we could well integrate them in the manuscript.
Reviewer 3 Report
Title: " Simultaneous predictions of chemical and phase equilibria in 2 systems with an esterification reaction using PC-SAFT "
The article studies the ability to predict the combined chemical equilibria and phase equilibria in multiple liquid-phase systems. In the article, the authors have tested the thermodynamic model Perturbed-Chain Statistical Associating Fluid Theory (PC-SAFT) for the modeling of chemical equilibrium in systems with liquid-liquid equilibria (LLE) and to develop an algorithmic approach to perform this task. This work presents a successfully designed and implemented in order to predict CPE in multiphase multicomponent systems. A good presentation of the results followed by discussions, conclusions and recommendations.
Overall, the paper is excellent for publication in the journal, after making the requested correction. I recommend this article for a publication in Molecules with a minor revision.
Comments
- In the Algorithmic approach:What about the tolerance ???? value.
- Give more interpretation to explain deviations from experimental data in presence of acetic acid in CE and LLE.
Author Response
1) In the Algorithmic approach: What about the tolerance ???? value.
Answer The tolerance ???? can in principle be chosen by the user of the algorithm, and depends on the tradeoff between required precision and numerical rund-off error that can happen when ???? is chosen to be very low. For this work, we used a value of ???? = 10-8, which we have reported now to the manuscript. Thanks for the comment.
2) Give more interpretation to explain deviations from experimental data in presence of acetic acid in CE and LLE.
Answer We added an explanation of why we think acetic acid is the main cause of the deviation between experimental and predicted CPE and extended the outlook of the paper addressing the requirement of a more phenomenological parametrization strategy that reflects the complex association behavior of mixtures with acetic acid.
Round 2
Reviewer 2 Report
1. Please indicate the source of a large number of formulas in the manuscript if they are not original.
2. It is mentioned in the manuscript that “The binary interaction parameters used in this work were in part retrieved from the literature”. Whether these binary interaction parameters use the same model?
3. For the beauty of the manuscript, it is suggested to modify the formula to center it.
4. It is suggested that authors add a flow chart of the algorithm structure in order to express the calculation steps of the algorithm more clearly.
5. It is suggested that authors test the algorithm based on his own experimental data rather than others' data.
6. The authors are requested to explain the meaning of each face in Figure 7 and Figure 8 in detail.
Author Response
Dear Reviewer, thanks for the valid points, we asnwered these point-by-point!
The manuscript with changes highlighted is attached.
REV: Please indicate the source of a large number of formulas in the manuscript if they are not original.
ANSWER: Thanks for the comment. The first part of the manuscript (basically encompassing equations 2.1 – 2.11) contains equations that are discussed in the specific (but not in the general) literature, so we introduce and explain their background there with the corresponding reference. We re-checked them and added some more references, whenever it was required. Equations related to the proposed algorithm (equations 2.12 – 2.17) were developed within this work and are therefore without reference. Equations A.1 – A.10 describing the employed model PC-SAFT can be found in the original publication(s) of PC-SAFT, but the references are included in there as well.
REV: It is mentioned in the manuscript that “The binary interaction parameters used in this work were in part retrieved from the literature”. Whether these binary interaction parameters use the same model?
ANSWER: Thanks for the question. Yes, the binary interaction parameters use the same model, the EoS PC‑SAFT, and were also determined in the past within our group.
REV:For the beauty of the manuscript, it is suggested to modify the formula to center it.
ANSWER We agree with the reviewer on the necessity to adapt the equations to the manuscript template. We centered the equations and added empty space before and after each equation to improve the readability of the manuscript.
REV:It is suggested that authors add a flow chart of the algorithm structure in order to express the calculation steps of the algorithm more clearly.
ANSWER: Now we included a flowchart which describes the computational step, performed within the algorithm, required to calculate the CPE. Thanks for the suggestion.
REV:It is suggested that authors test the algorithm based on his own experimental data rather than others' data.
ANSWER Thanks for the suggestion, but unfortunately, we don´t have own experimental data for the investigated systems actually. The purpose of this paper is to test the developed framework within PC-SAFT to represent the phase equilibrium of reactive LLE systems and to present an algorithmic approach for the efficient computation of the reactive phase diagrams. We didn´t aim at publishing new experimental results to fulfill missing data in the literature or to validate existing ones. We obtained the experimental data from the group of Toikka and of Prof. Hasse, two groups that are specialized in experimental determination of phase equilibria. Anyway, we plan to extend our scope to some more experiments in the future work!
REV: The authors are requested to explain the meaning of each face in Figure 7 and Figure 8 in detail.
ANSWER Figures 7 and 8 are so-called quaternary diagrams, which means that they represent the composition space (in general molar or weight fractions) of a 4-component system, including the pure components and the respective binary and ternary subsystems. For instance if, in Figure 7, 1-Hexanol is absent in the system, the quaternary diagram reduces to the ternary face Water-Acetic Acid-Hexyl Acetate, and any composition point without 1-Hexanol will then be only in the front face. The same holds if any of the other component is not present in the system. So, composition points containing all the 4 components will be entirely contained within the tetrahedron. If one component is not present, the composition point will lye on one face. If two components are absent, the corresponding composition point will be on one of the edges. If only one component is present, the composition point (the “pure‑component” point) will be on the respective vertex representing the pure component (for instance, in Figure 7, the point representing pure acetic acid is the upper vertex).
